# Potential Benefits of Seed Priming under Salt Stress Conditions on Physiological, and Biochemical Attributes of Micro-Tom Tomato Plants

**DOI:** 10.3390/plants12112187

**Published:** 2023-05-31

**Authors:** Nasratullah Habibi, Shafiqullah Aryan, Mohammad Wasif Amin, Atsushi Sanada, Naoki Terada, Kaihei Koshio

**Affiliations:** 1Graduate School of Agriculture, Tokyo University of Agriculture, 1-1-1 Sakuragaoka, Setagaya-ku, Tokyo 156-8502, Japan; a3sanada@nodai.ac.jp (A.S.); nt204361@nodai.ac.jp (N.T.); koshio@nodai.ac.jp (K.K.); 2Faculty of Agriculture, Balkh University, Balkh 1701, Afghanistan; 3Faculty of Agriculture, Nangarhar University, Nangarhar 2601, Afghanistan; shafiqaryan@gmail.com; 4Faculty of Agriculture, Parwan University, Parwan 1102, Afghanistan; wasifamin1991@gmail.com

**Keywords:** tomato, salinity, seed priming, photosynthesis, biochemical attributes, fruit quality

## Abstract

Pre-sowing seed priming is one of the methods used to improve the performance of tomato plants under salt stress, but its effect photosynthesis, yield, and quality have not yet been well investigated. This experiment aimed to alleviate the impact of sodium chloride stress on the photosynthesis parameters of tomato cv. Micro−Tom (a dwarf *Solanum lycopersicum* L.) plants exposed to salt stress conditions. Each treatment combination consisted of five different sodium chloride concentrations (0 mM, 50 mM, 100 mM, 150 mM, and 200 mM) and four priming treatments (0 MPa, −0.4 MPa, −0.8 MPa, and −1.2 MPa), with five replications. Microtome seeds were subjected to polyethylene glycol (PEG6000) treatments for 48 hours for priming, followed by germination on a moist filter paper, and then transferred to the germination bed after 24 h. Subsequently, the seedlings were transplanted into the Rockwool, and the salinity treatments were administered after a month. In our study salinity significantly affected tomato plants’ physiological and antioxidant attributes. Primed seeds produced plants that exhibited relatively better photosynthetic activity than those grown from unprimed seeds. Our findings indicated that priming doses of −0.8 MPa and −1.2 MPa were the most effective at stimulating tomato plant photosynthesis, and biochemical contents under salinity-related conditions. Moreover, primed plants demonstrated relatively superior fruit quality features such as fruit color, fruit Brix, sugars (glucose, fructose, and sucrose), organic acids, and vitamin C contents under salt stress, compared to non-primed plants. Furthermore, priming treatments significantly decreased the malondialdehyde, proline, and hydrogen peroxide content in plant leaves. Our results suggest that seed priming may be a long-term method for improving crop productivity and quality in challenging environments by enhancing the growth, physiological responses, and fruit quality attributes of Micro-Tom tomato plants under salt stress conditions.

## 1. Introduction

Tomatoes are widely recognized as an essential vegetable globally, including in China, the United States, Turkey, Italy, India, and Egypt [1]. They are a common ingredient in many meals worldwide [2]. Despite their popularity, tomato producers face various challenges, including constraints that limit tomato production, such as abiotic stress factors. Among these factors, salinity is one of the most significant issues in almost half of the world’s soils. Salt-affected soils, comprising 63.2 million hectares of topsoil (0–30 cm) and 120.0 million hectares of subsoil (30–100 cm), are formed due to concentrated salts and ions, such as ${Na}^{+}$, leading to the degradation of soil particles [3].

Salinity in soil poses a problem for most plants, and it is considered a significant factor causing injury to vegetable crops such as tomatoes on a global scale [4]. It negatively impacts several plant characteristics [5,6,7]. As a result, the weight of roots and shoots falls because it reduces nutrient absorption due to the build-up of harmful ions such ${Na}^{+}$ and ${Cl}^{-}$ [7]. In vegetable crops such as tomatoes, the growth parameters, including plant height and dry biomass, would be affected under salinity conditions [8]. Furthermore, salinity stress also impairs photosynthesis [9], which disrupts metabolic pathways and lowers turgor pressure [10], both of which result in high leaf surface temperatures and the closing of leaf stomata [11]. Exposure to high levels of salinity stress causes changes in transpiration, making it difficult for the plant to cool its leaves. This results in an increase in leaf surface temperature and harm to the leaf cells [12,13,14].

The effects of salt stress on tomato biochemical parameters such as sugars and organic acid, are well known. High salt stress conditions delay the fruit bearing in tomato plants, and furthermore fruits cannot reach to the full ripening stage as the ones under normal conditions [15,16]. Mild salt stress can increase the sugars, but high salt stress conditions decrease the sugars such as glucose, sucrose, and fructose [17,18], but adversely increase the accumulation of organic acids (citric acid and malic acid) in tomato fruits [19].

At present, scientists are exploring methods to mitigate the harmful impact of salinity on tomatoes or enhance certain mechanisms in them that enable adaptation to salinity-induced stress [20,21]. These mechanisms are influenced by several factors, including plant growth and physiological parameters, which vary across various stages of plant development [22,23,24,25]. Breeding and developing salt-tolerant varieties have been attempted but with limited success on local varieties. As a result, alternative and cost-effective methods have been proposed in the literature to enhance salt tolerance in plants before exposure to stress [26,27,28]. Seed priming is one such method that has been demonstrated to be effective in improving the traits of tomato plants under salt stress, using various agents such as polyethylene glycol (PEG6000) [29]. It has been also observed that osmo-priming of tomato seed with PEG6000 improves plant height and dry biomass during the vegetative stage [29,30,31]. Furthermore, priming is effective on some physiological parameters, such as photosynthetic rate, transpiration rate, and stomatal conductance [32]. Researchers found that seed priming with polyethylene glycol increases the SPAD value and chlorophyll content [32] compared to non-primed plants [33,34]. In addition, osmo-priming with polyethylene glycol mitigates the toxicity of the ions that accumulate in plant roots due to severe salt stress; by stabilizing the balance between cations and anions, polyethylene glycol diminishes the negative effects of $Na^{+}$ and $Cl^{-}$ ions accumulated in leaves due to sodium chloride salinity stress, by increasing the amount of $K^{+}$ ions [35,36]. Additionally, salt stress conditions can lead to a decrease in growth and photosynthetic activity, ultimately reducing fruit yield. However, pre-sowing seed priming can enhance antioxidant activity, thereby promoting a more active photosynthesis process in plants derived from primed seeds. Consequently, pre-sowing seed priming has the potential to increase fruit yield [37,38,39,40,41]. 

Furthermore, seed priming has a positive effect on aged seeds [42,43] to have better germination and produce stronger seedling with better physiological and biochemical performance [44,45,46,47]. This study not only assesses plants’ growth and physiological responses but also evaluates the quality attributes of the fruit. This aspect holds great significance as the quality of the final product is vital for both commercial purposes and human consumption. The study’s outcomes offer valuable insights into the potential advantages of pre-sowing seed priming in enhancing both crop yield and fruit quality, particularly in stressed environmental conditions. The effects of pre-sowing seed priming with polyethylene glycol on tomato plant photosynthesis, yield, and fruit quality under salt stress conditions have not been reported previously. Pre-sowing seed priming methods can boost crop yield, enhance crop quality, and reduce plant vulnerability to abiotic stressors, such as salt stress. Moreover, as the global population grows and cultivable land becomes limited, it is increasingly crucial to maximize crop productivity with limited resources. In light of salinity’s detrimental effects, the current study was carried out to (a) assess salinity’s effects on tomato growth and physiological parameters, (b) evaluate the comparative effects of salinity and polyethylene glycol on tomato growth, physiological, and biochemical parameters, and (c) to investigate the interaction between polyethylene glycol and salinity.

## 2. Results

### 2.1. Growth Parameters

Salinity had a considerable impact on all growth indicators, and it was shown that these characteristics declined as the salinity concentration rose. In terms of plant height, there was no discernible difference between priming treatments at 0 MPa and control conditions; however, at 50 mM salinity, treatments at −0.8 MPa and −1.2 MPa had higher plant height (12.70 cm and 14.12 cm) than unprimed plants (12.53 cm), with −1.2 MPa being the best. All priming treatments increased plant height over control in salinities of 100 mM, 150 mM, and 200 mM, although the best treatments had salinities of −1.2 MPa, −0.8 MPa, and −0.4 MPa, respectively (Table 1). Generally, plant height was reduced by salinity, but priming treatments had a positive effect, and the interaction between salinity and priming was strongly negative. Moreover, a significant interaction (*p* < 0.05) between salinity and PEG existed. Similarly, salinity led to a decrease in root length, while PEG treatments resulted in an increase. Normal conditions (S0) did not significantly differ between priming treatments and control, whereas salinity conditions did significantly differ between priming treatments and no priming, with −1.2 MPa being the most effective priming treatment for root length. In comparison to control conditions, plants exposed to salinity produced fewer leaves decreasing from 26.67 in control condition to 16.11 in 200 mM salinity. However, plants exposed to PEG treatments produced noticeably more leaves than those exposed to non-primed treatments (0 MPa), and the interactions between salinity and PEG were significant in terms of the number of leaves produced per plant, leaf length, and leaf area, respectively. Under control and 50 mM salinity, priming has a negligible effect on the number of leaves per plant, but it has an obvious effect at 100 mM, 150 mM, and 200 mM. Salinity considerably reduced the number of branches per plant, whereas priming treatments enhanced it, and there was a significant interaction between salinity and PEG (*p* < 0.05). Although there was no apparent variance between priming treatments and controls under normal circumstances, this was not the case when salt stress was present. 

### 2.2. Plant Biomass

Table 2 demonstrates that decreasing PEG levels and raising salinity have an adverse impact on plant growth. All three plant parts’ FW and DW values were highest for plants grown under the S0P0 treatment and lowest for plants cultivated under the S4P2 treatment. The effect of the S and P factors is also evident from the significant differences between the treatments. For instance, seed priming has an advantageous effect on plant turgor and growth because the FW and DW of shoots and leaves in the S0P0 treatment are much higher than those in the S0P1 and S0P2 treatments. Similar to this, the FW and DW of roots in the S0P0 treatment are significantly higher than those in all other treatments except S0P3. Salinity had a negative impact on root fresh weight (*p* < 0.001), but −1.2 MPa priming treatments were only successful at 0 mM and 100 mM salinities. These treatments were ineffective at 50 mM, 150 mM, and 200 mM salinities. Although there is no apparent distinction between primed and non-primed plants under salt stress, −1.2 MPa is preferable to 0 MPa under 0 mM salinity. In salinities of 0 mM and 200 mM, priming had no effect on the shoot’s fresh weight, but it was effective at −0.4 MPa and −1.2 MPa, which were superior to 0 MPa at 50 mM and 150 mM salinities, respectively. Under the control condition (no salinity), priming had a detrimental impact on the shoot’s dry weight, but under salt stress, a priming treatment with a pressure of −0.4 MPa was more desirable than a treatment of 0 MPa. The priming treatments did have a negative impact on the fresh weight of the leaves. However, −0.8 MPa and −1.2 MPa were much better under 0 mM, and 50 mM salinity, whilst under 200 mM salinity −0.4 MPa and −1.2 MPa priming treatments were substantially superior to 0 MPa. Additionally, there was virtually no difference between priming treatments under 100 mM and 150 mM salinity treatments for leaf dry weight.

The results presented in Table 3 indicate that plant growth is significantly influenced by salt and priming levels. Higher salinity and lower PEC concentrations generally lead to decreased FW and DW values in roots, shoots, and leaves. Our study also found that salinity considerably impacted the fresh and dry weight of leaves. Additionally, PEG was found to be effective in enhancing both the fresh and dry weight of plants. Although it has been reported that seed priming can induce plant dry weight under salt stress conditions, the outcomes of this experiment indicated improvement in root biomass too.

### 2.3. Photosynthetic Parameters

The mean values of photosynthetic parameters as affected by the treatment combinations of salinity (S) and PEG (P) levels are shown in Table 3. Generally, the examined (or applied) treatments were found to affect significantly all the measured growth and photosynthetic attributes. The augmented levels of salinity in the absence of PEG (S1P0, S2P0, S3P0, and S4P0) was associated with liner significant decrease in photosynthetic rate, transpiration rate, and stomatal conductance, whereas the highest level of salinity (S4P0) gave the highest mean value of leaf surface temperature. Despite this, the greatest PEG concentration in the absence of salt (S0P3) was shown to improve the physiological characteristics of photosynthetic rate, transpiration rate, and stomatal conductance, whereas the same treatment led to a considerable drop in leaf surface temperature when compared to the control. Additionally, under conditions of the highest salinity, the highest concentration of PEG (S4P3) was discovered to dramatically lessen the detrimental impact of salt on photosynthetic characteristics as compared to the S4P0 treatment. The study shows that the main effects of salinity (S) and PEG (P) were both significant for all parameters, and the interaction between S and P was also significant. The highest values for net photosynthesis, transpiration rate, and stomatal conductance were observed in the S0P3 treatment (Table 3).

### 2.4. SPAD Value

Relative chlorophyll content (SPAD value) was negatively affected by salinity. The results revealed that seed priming with PEG6000 significantly affected the SPAD value of tomato plants under saline environmental conditions. Seed-primed plants exhibited significantly higher SPAD values compared to non-primed plants. For instance, during vegetative growth, a −1.2 MPa application of PEG6000 enhanced the SPAD value. Results revealed that salinity and PEG had significant effects, although in opposing directions. The variations were notable. Therefore, it can be said that seed priming enhances the SPAD value in tomato plant leaves that are under salt stress (Figure 1). The results indicate that priming is a useful technique to increase the chlorophyll levels in tomato plants when they are exposed to unfavorable salt stress conditions. This improvement in chlorophyll content is a sign of enhanced photosynthetic efficiency and better growth of the plants.

### 2.5. Fruit Yield Attributes

The effects of salinity treatments (S0–S4) and four levels of PEG6000 (P0–P3) on a particular tomato plant’s fruit size, weight, number of fruits per plant, and yield per plant are showed in Table 4. Under stressful conditions with salt, tomato fruit size responded favorably to seed priming. Salt stress negatively impacted fruit growth, yet seed priming had a large positive impact. The smallest fruit size was produced by the S4P0 treatment, and the biggest fruit size was formed by the S1P3 treatment. Salt exposure lowered fruit weight, although treatments for seed priming had no obvious effect on this parameter. For instance, the S0P0 treatment resulted in the heaviest fruit (9.04 g), whereas the S4P3 treatment led to the lightest fruit (1.18 g). Seed priming was effective on the fruit size of tomatoes under salt stress conditions, and it was decreased as an effect of salt stress, but seed priming could significantly enhance the fruit size. The S4P0 treatment resulted in the smallest fruit size (11.2 mm), while the S1P3 treatment with a fruit size of 20.7 mm produced the largest fruits (Table 4). Similarly, fruit weight was decreased by salt stress, but seed priming treatments did not bear a significant effect on this parameter. The number of fruits per plant was higher in seed priming treatments than in control ones. S0P2 and S0P3 treatments produced the highest number of fruits per plant, while S3P0 and S4P0 produced the lowest. Seed priming treatments can substantially increase fruit yield per plant due to larger fruit size and higher fruit count per plant. Among the treatments, S0P3 resulted in the highest yield per plant, while S4P0 gave the lowest. The interaction between salt treatment and PEG content significantly influenced all four assessed fruit attributes.

### 2.6. Fruit Quality Parameters

Table 5 demonstrates how fruit quality parameters are affected by the interaction of different seed priming methods (P) and salinity concentrations (S). The result of the trial shows that treatments S2P2 and S3P3 had the highest fruit color values, while treatment S1P0 had the lowest values, indicating that the tomato fruits were redder in the treatments in which the priming was modified without salt as well as in those whereas the amount of salt was adjusted with high PEG6000 concentrations. Nevertheless, the fruits in the treatments within which the seeds were not primed and/or the amount of PEG was reduced, and the amount of salt climbed took longer to mature. This result implies that the salt stress level and its relationship with seed priming moderately impacted fruit color. The findings demonstrate that treatments S4P2, S4P3, and S4P1 exhibited the greatest fruit Brix (%), in contrast to S0P0 and S1P0 showed the lowest values. Due to the smaller size of the fruit in low salinity compared to the control, fruit Brix (%) and three sugars, namely glucose, fructose, and sucrose, tended to be higher, while the number of sugars decreased at elevated salt concentrations. In this regard, treatment S1P1 possesses the most rigorous, and treatment S4P0 has resulted in minimal glucose levels. Likewise, the amount of fructose was found to be most prevalent in treatment S1P2, S1P1, followed by S1P0, and the poorest in treatment S4P3. Treatments S0P3 and S0P1 possessed the biggest and at their lowest sucrose levels, respectively. The treatments S0P3 and S0P1 have the highest and lowest sucrose concentrations. 

The sugar level was increased once again due to the priming treatments, even though salinity conditions typically lead to higher quantities of organic acids, such as citric acid, malic acid, and vitamin C. Treatment S4P0 had the highest concentration of citric acid, while treatment S0P3 had the lowest. The treatments S4P0 and S4P1 resulted in the highest concentrations of malic acid, while S0P1 and S0P2 showed the least amount. Similarly, treatments S4P2 and S4P3 had the highest levels of vitamin C, while S0P2 and S0P3 had the lowest levels. The table shows that the main effects of S and P and their interaction effects are statistically significant for fruit quality characteristics. Since sugars and organic acids have an inverse relationship, a decrease in sugar content leads to an increase in organic acid proportion, resulting in a sour taste for the fruit.

### 2.7. Fruit Ethylene Production

The fruit ethylene production analysis findings indicate a notable difference between the priming treatments and the control group in each salinity level. This suggests that seed priming treatments can expedite fruit maturity, even though salt stress may cause a delay in the process, as shown in Figure 2.

### 2.8. Leaf Biochemical Content

Lipid peroxidation is resulted when plants face stress, therefore, some biochemicals will be released be plant tissue in response to stress conditions. Salt stress induced lipid peroxidation as high electrolyte leakage in leaves, and simultaneously MDA, proline, and hydrogen peroxide were also increased. However, the plants under seed priming treatments had a lower electrolyte leakage, MDA, proline, and hydrogen peroxide content in their leaves compared to non-primed ones. Furthermore, the results revealed a significant interaction (*p* < 0.01) between salt stress treatments and seed priming treatments (Table 6). 

Leaf electrolyte leakage was measured as an injury effect of salt stress, and the results showed that electrolyte leakage was incredibly induced by salinity increment. Under severe salt stress condition (200 mM) a high electrolyte leakage (50.36%) was observed which is very critical during the reproductive stage (flowering and fruit-bearing). However, a decrease of more than 10% percent in electrolyte leakage of the plants under seed priming treatments was observed which is showing the better performance of the plants compared to the control (Table 6).

The data of leaf nutrient analysis show a negative effect of salt stress on the accumulation of the nutrient elements in plant leaf tissues; however, the seed priming treatments were effective too. Plants subjected to salt stress showed a high accumulation of sodium ions in their leaves. This increment was in parallel with the accumulation of calcium and zinc ions, but the amount of potassium, phosphorus, magnesium, and iron decreased which shows the antagonistic role of sodium with the elements. Seed priming could enhance the accumulation of beneficial elements such as potassium, phosphorus, magnesium, and iron (Table 7). Furthermore, the ratio between sodium and potassium ions was also diminished under priming treatments compared to the non-primed plants. 

### 2.9. Principal Component Analysis

Principal component analysis of the photosynthetic parameters (PC1 = 47.8% and PC2 = 26.2%) explains 74.0% (PC1 and PC2) of the variance in all the data, hence PC1 and PC2 are sufficient to contribute the variance of all the data. According to PC1, there is a strong positive correlation between stomatal conductance and transpiration rate, which suggests that when more stomata open, the transpiration rate will increase as a result. Additionally, as there is a positive correlation between photosynthetic rate, stomatal conductance, and transpiration, it can be inferred that whenever one of the aforementioned attributes is impacted, the other two are going to be affected as well (Figure 3). In contrast, the remaining photosynthetic traits exhibited a negative correlation with leaf surface temperature, indicating that an increase in the leaf surface temperature is likely to result in a decrease in the other photosynthetic features. The PCA results revealed that the plants under seed priming treatments specifically −0.8 MPa and −1.2 MPa had higher leaf area, photosynthetic rate, and chlorophyll content and the stomata were open which led to more transpiration rate. However, the plants grown in non-primed treatments (S1P0, S2P0, and S3P0) had lower photosynthetic and closer stomata with lower transpiration rates which caused an increase in leaf surface temperature.

In principal component analysis of the biochemical parameters, PC1 explains 83.4% of the variance in all the data, hence PC1 is sufficient to contribute to the variance of all the data. This shows that sugars (glucose, fructose, sucrose) have a strong correlation with potassium. Furthermore, the analysis shows that sodium and calcium have a negative correlation with potassium and sugars (Figure 4).

### 2.10. Regression Analysis

Electrolyte leakage, an essential component of plants that indicates the level of cell damage, directly impacts photosynthetic parameters. The presence of salt causes the foliage to discharge electrolytes more frequently, indicating an increase in membrane permeability and the deterioration of leaf tissues. Salinity also affects photosynthetic features, and our research found a significant correlation between electrolyte leakage and photosynthetic parameters. Regression analysis revealed a positive correlation between electrolyte leakage and leaf surface temperature, indicating that plant cell membranes become more permeable under stress, potentially leading to increased electrolyte leakage. Meanwhile, the plant’s ability to regulate internal temperature through transpiration may be impaired, resulting in an increase in leaf surface temperature. Conversely, electrolyte leakage was negatively correlated with photosynthetic rate, transpiration rate, and stomatal conductance, indicating that the plant can maintain the integrity of its membranes and its water-use efficiency under stressful conditions, as shown in Figure 5.

### 2.11. Correlation Analysis

According to correlation analysis, there was a significant relationship between photosynthesis and fruit quality parameters such as glucose, fructose, and sucrose. This suggests that plants with higher levels of photosynthesis we better in sugar content and with low acidity. This demonstrates how seed priming with PEG6000 may optimize a plant’s ability to increase photosynthesis and imply improved fruit quality performance under saline environmental conditions. More importantly, fruits per plant, yield per plant, fruit size, glucose, stomatal conductance, and transpiration rate all showed a strong positive correlation with photosynthetic rate; however, photosynthetic rate showed a negative correlation with citric acid, malic acid, and leaf surface temperature. As a result of this, the plants grown from primed seeds are capable of withstanding stress simply, because it develops longer roots, which at first promote water absorption, increased transpiration, and decreases leaf surface temperature as well as prevents cell damage due to electrolyte leakage (Figure 6). Furthermore, primed plants have increased stomatal conductance, which results in higher gas exchange rates in the leaves, promoting photosynthesis. Considering fruit quality, fruit glucose content had a significant correlation with the photosynthetic rate (r = 0.64), transpiration rate (r = 0.84), stomatal conductance (r = 0.75), potassium (r = 0.79), and phosphorus (r = 0.89). Adversely, the photosynthetic rate had a negative significant correlation with hydrogen peroxide (r = −0.75), proline (r = −0.78), and MDA (r = −0.78), which can be concluded that primed plants had active photosynthesis which affected the lipid peroxidation and therefore, the less amount of oxidative stress biochemicals were produced in leaves. In addition, in the case of nutrients, sodium had a negative correlation with photosynthesis-related parameters and fruit sugars. It proves that when plants are subjected to salt stress, sodium accumulates in the leaves and causes a decline in photosynthesis and carbohydrate formation. The data also show a positive correlation of potassium, phosphorus, and iron with photosynthesis and fruit quality. Seed priming treatments caused a decrease in the accumulation of sodium and calcium and induced the accumulation of potassium, phosphorus, and iron. Therefore, it can be concluded that seed priming could enhance the antioxidant capacity of the plants to combat salt stress (Figure 6).

## 3. Discussion

### 3.1. Growth Parameters

It has been reported that growth parameters including plant height, root length, and leaf size are affected by salinity stress, which is parallel to our result, with a different in salt treatment concentration. Plant height in severe salinity was badly affected and some of the plants could not survive [4,5]. Furthermore, it has also been confirmed that all morphological parameters in tomato plants including root length [5] number of leaves, leaf length, leaf width, and number of branches affected by salinity [8], and reported 18 leaves per plant as the maximum number of leaves and 8 leaves per plant as the minimum number of leaves per plant while in our experiment, 23 was the maximum number of leaves per plant and 16 was the minimum number of leaves per plant. However, it has been found that the parameters especially plant height and dry matter was improved in tomato plants with primed seeds [17,21,29], which is the same as our result but the difference between their study and ours is the salt type, the variety of tomato, and we examined physiological responses of the tomato while primed and non-primed. 

Root fresh and dry weights were significantly decreased by salinity, while PEG was significantly effective in 50 mM and 100 mM salinity, but PEG was not significantly effective in 150 mM and 200 mM salinity. The interaction was significant between salinity and PEG in root fresh weight and root dry weight, respectively. Similarly to our results, other authors [29,30] also stated that salinity decreases dry matter in the tomato; however, seed pre-sowing priming with PEG6000 can improve dry weight. Shoot fresh weight decreased in salinity and PEG was effective and their interaction between them was *p* < 0.005, but shoot dry weight decreased significantly under salinity and significantly increased under priming treatments, and −0.4 MPa was the best PEG treatments among all, and the interaction between salinity and PEG was *p* < 0.001. The results from the previous study by other authors [30] also confirm that shoot fresh weight was increased under PEG treatments and the plants with pre-sowing priming with PEG6000 produced 23% more shoot fresh weight than the ones in control, and they concluded −0.75 MPa PEG as effective treatment which is near −0.8 MPa that we used in the current study. However, there is no previous study about the effect of seed priming with PEG6000 on the fresh weight of leaves and roots in tomato plants under sodium chloride stress. 

### 3.2. Photosynthesis Attributes

Seed priming was found to be improving on photosynthetic parameters of tomato plants. It was observed that seed priming could induce photosynthetic rate, and transpiration rate [48]. It is reported that photosynthetic pigments which are the important factors for leading the photosynthesis process to be efficient were enhanced in plants that were derived from the primed seeds with PEG6000 in mung bean (*Vigna radiata* L.). It has also been reported that seed priming with 0.75% (weight/volume) potassium nitrate for 24 h at 25 °C improved photosynthetic in tomato plants [13,14,48]. Moreover, it has been found that seed priming could improve growth, photosynthesis, and cucumber while plants were growing under ambient conditions. The authors reported that gibberellic acid and potassium nitrate were effective on net photosynthetic activity in cucumber leaves [49]. It has also been proved that pre-sowing seed priming with PEG could induce photosynthetic rate in rice seedlings under water stress [50]. A study on the effects of seed priming treatments on the germination and development of two rapeseed (*Brassica napus* L.) varieties under the co-influence of low temperature and drought reported that seed priming with abscisic acid and gibberellic acid could significantly induce the photosynthetic rate and transpiration rate compared to the non-primed plants [31,51]. Seed priming with PEG6000 was found to be efficient on increasing the amount of chlorophyll content of tomatoes under salt stress which is a good indicator for higher photosynthesis rate [14,52]. Previous studies have explored the impact of various priming agents on tomato physiological parameters in response to salt stress, and our study agrees with Lei et al. [48,53] but our findings indicate some beneficial physiological and biochemical improvement by seed priming with polyethylene glycol (PEG6000) on tomato under sodium chloride salt stress. Therefore, our study is novel and presents new findings compared to previous research. Our findings demonstrate that salt stress caused by sodium chloride reduces tomato photosynthesis due to increased cell injury caused by electrolyte leakage and an increase in leaf surface temperature, which negatively impacts the photosynthesis rate. However, our study found that seed priming with PEG6000 was highly effective in increasing the photosynthesis rate while also reducing leaf surface temperature and electrolyte leakage. Our results confirm the idea in the results of Moaaz Ali et al. [13]. However, the priming agent was not the same. According to Moaaz Ali et al. [13], seed priming with potassium nitrate can improve plant performance under stress conditions, which might be due to a fertilization role or due to the availability of nitrate and potassium for seeds to grow faster, but polyethylene glycol accelerates the activation phase in seeds which facilitates the usage of nutrients contents, and the seeds can develop vigorous seedlings. Therefore, the seedlings with more leaves can absorb more sunlight, leading to higher photosynthesis. It is also reported that priming increases the SPAD value and chlorophyll content of the tomato leaves [22,41], but the authors used distilled water, NaCl, and KNO_3_ as priming agents while we used PEG6000 as priming agent and also the difference is that they had 40.92 as maximum SPAD value that was under KNO_3_ priming and 22.56 as minimum SPAD value that was produced in control plants, but in our study, we found that priming with PEG6000 can increase the SPAD value to 63, and there is a 10–20% difference in SPAD value of primed and non-primed plants.

Furthermore, photosynthetic pigments which researchers reported to be increased under priming treatment can also be a good factor for higher photosynthesis; however, in the current study we have not checked the amount of pigmentation in leaves. Moreover, we confirm the result of Li and Zhang [49], because when plants face salt stress, normally experience drought stress due to less water absorption by roots, therefore, strong roots of plants that their seeds were primed with PEG may have more water use efficiency which could increase the drought tolerance in plant and stronger plants may have better photosynthesis [54].

### 3.3. Fruit Quality and Biochemical Attributes

Fruit yield and fruit quality attributes including fruit sugars (glucose, sucrose, fructose) and organic acids (citric acid and malic acid) were evaluated by the effect of seed priming treatments and salt stress conditions which is not previously reported. Although reports are showing the negative effects of salt stress on yield [33,37,38,39] and the sugar contents of tomato fruits. It was reported by Moaz Ali et al. that potassium nitrate could induce the total soluble solids (Brix) and phenolics through seed priming [40], but our study findings explain the increment in fruit’s sugars and a decrement in organic acids which are the key factor for perishable fruit in the market. 

Vitamin C is one of the antioxidants which declines the negative effects of lipid peroxidation that is caused by salt stress [55]. This parameter is expected to be decreased as electrolyte leakage [55] increases. In our experiment, there was a significant decrease in vitamin C content with the increment of salt stress, which is like the result reported in [56], but we could enhance the plant performance to induce the vitamin C by seed priming treatments which can be a good weapon for the plant to defense against salt stress. Furthermore, it has been reported that MDA, proline, and hydrogen peroxide are produced as a response to salt stress conditions. We also found in the current study that as the salt stress becomes severe, the production of MDA, proline, and hydrogen peroxide increases, and our results reveal that seed priming treatments could significantly decrease the production of the mentioned chemicals which shows the enhancement in plant tolerance to salt stress conditions. The accumulation of nutrients in leaves of tomato plants subjected to salt stress was reported to be increased by stress severity and duration and also that the accumulation of sodium [57] and calcium will increase, while there will be a significant decrease in the accumulation of potassium, phosphorus, and other important elements which react as antioxidants and decrease the injury level caused by lipid peroxidation. However, still, there is lack of research on improving fruit quality and antioxidant activity of tomato plants under severe salt stress conditions through seed priming; therefore, our findings are novel results, and thus seed priming could be an easy approach for farmers and tomato producers to increase the yield and fruit quality of their products. Moreover, our findings could lead to further research and more clarification in this regard.

## 4. Materials and Methods

### 4.1. Experimental Site and Design

This experiment was conducted at the Greenhouse Laboratory of Tropical Horticulture Science, Department of International Agricultural Development, Tokyo University of Agriculture (Setagaya Campus, Tokyo, Japan). Micro-Tom (wild type) seeds were employed as the plant material in the factorial completely randomized design (CRD) experiment, which contained five salinity treatments and four priming treatments. Each combination had ten replications, and one plant was considered as a replication. In total, 200 plants were used for measuring the parameters.

### 4.2. Application of Treatments

Polyethylene glycol (PEG6000) (Cica-Reagent, Tokyo, Japan) [58] was used to prime the seed before cultivation and the seeds were soaked in the solution for 48 h. The seeds were then thoroughly dried by being washed three times with clean water and then left on a table for three hours. The seeds were weighed after drying to approximate their initial weight. This experiment involved four priming treatments: −0 MPa, −0.4 MPa, −0.8 MPa, and −1.2 MPa. To determine how much PEG6000 should be dissolved in pure water to create priming treatments, the following formula was employed [17,18,20,59]:(1)$$OP=\left(-1.18\times 10^{-2}\right)\times C-\left(1.18\times 10^{-4}\right)\times C+\left(2.67\times 10^{-4}\right)\times C\times T+\left(8.39\times 10^{-7}\right)\times C^{2}T$$

In Equation (1), OP stands for osmotic pressure, the temperature is indicated by T, and PEG concentration is represented by C.

To apply salinity stress, sodium chloride was utilized in this experiment at five different concentrations: Control, 0 mM, 50 mM, 100 mM, 150 mM, and 200 mM. The current study is a two-dimensional experiment when salinity stress and priming treatments are considered. Table 8 lists 20 treatments that combine salinity and priming.

At the time of the seedling’s transplantation to the plant bed, salinity was introduced within irrigation water, and an irrigation period of three days was conducted when the seedlings were at the 3–4 leaf stage. The salinity treatments were applied until fruit harvest within irrigation water. The irrigation water was obtained from tap water, and its pH level was adjusted to 5.8–6 using sodium hydroxide and hydrochloric acid. Additionally, NaCl was added to the irrigation water. The plants were irrigated with 100 mL of saline water each time and for avoiding the salt concentration, every time the irrigation water was prepared freshly, and the plants in each treatment were irrigated individually.

### 4.3. Growth Parameters

Throughout the transplanting stage to the flowering stage, six measurements of the plant’s height were taken every week. The growth characteristics were evaluated six weeks after transplantation. In each treatment, 10 plants were chosen to determine the number of leaves per plant, and the leaves were counted for further analysis. A ruler was utilized to measure the length of leaves, and at the sixth week after transplanting, the number of branches per plant was counted. In addition, the ImageJ software was used to measure leaf area after scanning the leaves with a digital scanner (EPSON DS-G2000, Tokyo, Japan). Ten random leaves were selected from different plants in each treatment for leaf area measurement, and the data were recorded for further analysis. Furthermore, a digital scale was used to directly measure the fresh weight of shoots, leaves, and roots. To measure dry weight, the samples were placed in an oven at 70 °C for 24 hours, and then the dry weight of shoots, roots, and leaves was measured using a digital scale. There were six replications for each plant biomass parameter. 

Root samples were taken from ten plants for root length measurement. Root samples were taken once while the fruits were harvested. A ruler was used to measure the root length, and a digital balance was used to measure the fresh weight right away. The roots were then stored individually in envelopes and dried thoroughly for 48 hours in an oven at 60 °C. A digital balance was used to measure the root dry weight after thoroughly drying the roots.

### 4.4. Photosynthetic Parameters

The LCi-SD Portable Photosynthesis System (ADC Bioscientific, Hoddesdon, UK) was used to measure the photosynthetic rate (P_n_: μmol CO_2_ m^−2^ s^−1^), transpiration rate (E: μmol CO_2_ m^−2^ s^−1^), stomatal conductance (gs: mol CO_2_ m^−2^ s^−1^), and leaf surface temperature (LT: °C) two weeks after applying salinity treatment. The measurements were taken from ten plants for each treatment. The leaf chlorophyll content was also measured using a SPAD value meter (SPAD-502 Plus; Spectrum Technologies, Aurora, IL, USA).

### 4.5. Fruit Yield Parameters

Once all the plants’ fruits had reached full ripeness, the number of fruits produced per plant, fruit weight (g), fruit size (fruit diameter in mm), and total fruit production (g) per treatment were recorded.

### 4.6. Quantification of Biochemical Parameters

Ten fruits per treatment (one fruit per plant) were used for the quantification of fruit sugars, including glucose, sucrose, and fructose, as well as organic acids such as malic and citric acids. The quantification was carried out using HPLC with a differential refractometer (RID-10A, Shimadzu, Tokyo, Japan) on a KS-801 (8.0 mmI.D. × 300 mm, Shodex, Tokyo, Japan) and KS-G (6.0 mmI.D. × 50 mm) column, as mentioned in reference [11]. Additionally, the production of ethylene in the fruit was evaluated to confirm its ripeness, and gas chromatography was utilized to measure the amount of ethylene produced, and the values were transformed as $nL\times g^{-1}\times h^{-1}$, as cited in the same reference. Moreover, fruit color was measured using a handy colorimeter (NR-3000, Nippon Denshoku Ind., Ltd., Tokyo, Japan), the * a value color was considered in this experiment as it expresses the red color. Vitamin C was measured based on the method described by the Association of Official Analytical Chemistry [39]. Ten grams of cut fresh fruit was added into 10 mL of 5% meta phosphoric acid and crushed well using a pestle and mortar and 2 mL of mixed juice were taken in a tube and centrifuged (TOMY MX-307 high-speed refrigerated microcentrifuge, Tokyo, Japan) for 10 min in 9000 RPM at 25 °C, and then vitamin C was measured using reflectometer (RQ-flex Plus, Merck Inc., Darmstadt, Germany) handy meter. Ten randomly selected fruits were used for measuring vitamin C in each treatment and one fruit was considered as a replication. The vitamin C amount was measured three times in each replication and the final vitamin C content was calculated using the formula below:(2)$$Vitamin\;C=\frac{SW+CW}{SW}\times AR\times 10^{-1}$$

In Formula (2), SW is the fruit sample weight, CW is the chemical (meta phosphoric acid) weight, and AR is the average of three times the machine reading. 

By determining the number of electrons that were leaking from the leaves, electrolyte leakage was assessed to determine the extent of leaf damage [52]. Ten random leaves in each treatment were taken for measuring electrolyte leakage and each leaf was considered as a replication. An electrical conductivity meter (LAQUATWIN-S070, Horiba Scientific Ltd., Kyoto, Japan) was used to measure the electric conductivity (EC) of leaf cuttings that were made using a 1-cm diameter stainless steel cork-borer and stored in pure water in 2-ml tubes under room temperature (25 ± 1 °C) for half an hour. 

Malondialdehyde (MDA) was measured from ten random leaves of different plants in each individual treatment, the procedure followed the protocol as previously described in [60], with some changes. Extracts were mixed with 0.5% thiobarbituric acid (TBA) prepared in 20% trichloroacetic acid (TCA) (or with 20% TCA without TBA for the controls), and then incubated at 95 °C for 20 min, cooled on ice to stop the reaction and centrifuged at 13,300× *g* for 10 min at 4 °C. The absorbance was measured at 440, 600, and 532 nm. The MDA concentrations were calculated using the equations included in [61,62]. MDA contents were expressed as $nmol\;g^{-1}\;DW$.

Leaf proline content was also measured from ten randomly selected leaves of different plants in a treatment. Proline content was analyzed by the modified procedure of Gharsallah et al. [62]. Approximately 100 mg of the crushed leaves were homogenized with 250 μL each of methanol (analytical reagent grade) and chloroform (analytical reagent grade) and heated in a water bath for 5 min at 37 °C. Ribitol solution (50 μL) and pure water (175 μL) were added, and vortexed. All samples were centrifuged for 10 min at 140 × 100 RPM, and 80 μL of the supernatant from each sample was vaporized using a vaporizer and placed in a freeze dryer for 24 h. A solution of 20 mg of methoxamine hydrochloride and 1 mL of pyridine was prepared and 40 μL of the prepared solution was added to each sample and was heated at 37 °C for 90 min. Finally, 50 μL of MSTFA (N-Methyl-N-trimethylsilyltrifluoroacetamide) solution was added to each sample and analyzed for proline using a gas chromatograph-mass spectrophotometer (GCMS-QP2010 Plus Shimadzu, Japan). The amount of free proline was quantified using a standard curve and expressed as $\mu mole\;g^{-1}$ tissue fresh weight.

Hydrogen peroxide (H_2_O_2_) was measured from 50 mg of fresh leaf material extracted with a 0.1% (*w*/*v*) trichloroacetic acid (TCA) solution, followed by centrifuging the extract. The supernatant was mixed (1:1) with potassium phosphate buffer (10 mM, pH 7.0) and (1:2) KI (1 M). The absorbance was measured at 390 nm [62]. Concentrations were calculated against an H_2_O_2_ standard calibration curve and expressed as $\mu mol\;g^{-1}\;DW$. 

Leaf nutrient elements such as sodium (Na), potassium (K), phosphorus (P), calcium (Ca), magnesium (mg), iron (Fe), and zinc (Zn) were analyzed using the method described by [57]. Tomato leaves were dried in an oven under 80 °C for 48 h and then crushed using a grinder. Of the samples, 0.1 g of each was kept inside 5 mL of nitric acid for 5 h, and later the nutrient concentration was analyzed using a spectrophotometer (Varian spectra AA 220, Varian, Palo Alto, CA, USA). The values presented as $mg\times g^{-1}\;DW$.

### 4.7. Statistical Analysis

R 3.6.2 statistical software was used to analyze the data, while Python 3.7.4 Jupyter Notebook “https://api.anaconda.org (accessed on 10 November 2022)” was used to visualize the data. Analysis of variance (two-way ANOVA) at a 5% level of significance, Tukey’s test, Pearson’s correlation analysis, regression analysis, and principal component analysis (PCA) were performed to fully comprehend the findings.

## 5. Conclusions

Tomato plant photosynthesis is impeded by salinity, resulting in elevated electrolyte leakage and cellular damage. Additionally, salinity affects tomato plant growth parameters, such as plant height, leaf area, and leaf number. Nevertheless, priming tomato seeds with polyethylene glycol (PEG6000) before sowing strengthens the adaptation mechanism in seeds, promoting pre-tolerance in plants at the early stages of growth following planting. As a result, primed seeds produced healthier and faster-growing plants, with the additional benefit of exhibiting more active photosynthesis than unprimed seedlings. This study extensively examines the use of seed priming to overcome the salt barrier and improve the development, physiology, and fruit quality of Micro-Tom tomatoes. The results demonstrate that this method has the potential to considerably increase crop yields while enhancing the plants’ salt stress resistance. This discovery has profound implications for agricultural production, particularly in highly salinized regions where farmers struggle to grow resilient crops. By making seed priming a common practice, we may help agricultural systems’ production, increase food security, and diminish the environmental impact of farming. Researchers can employ the findings of this study as a benchmark for similar investigations on different plants. Future research endeavors are likely to concentrate on the impacts of pre-sowing seed priming on plant growth and stress tolerance.

## Figures and Tables

**Figure 1 plants-12-02187-f001:**
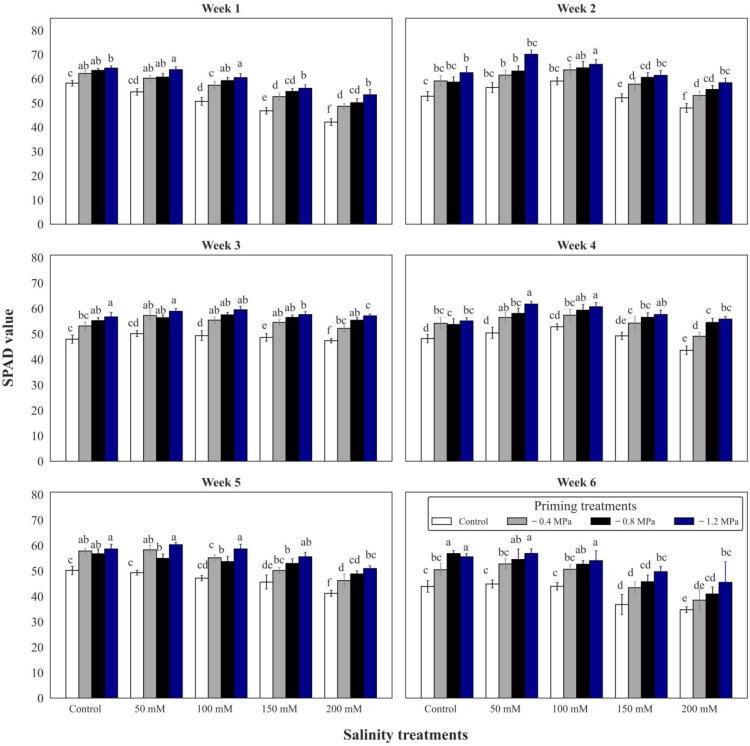
SPAD value during six weeks of vegetative growth, while the plants were at two months of age. The bars with the same alphabetical letters for each treatment and week did not differ significantly, based on the Tukey test at 5%.

**Figure 2 plants-12-02187-f002:**
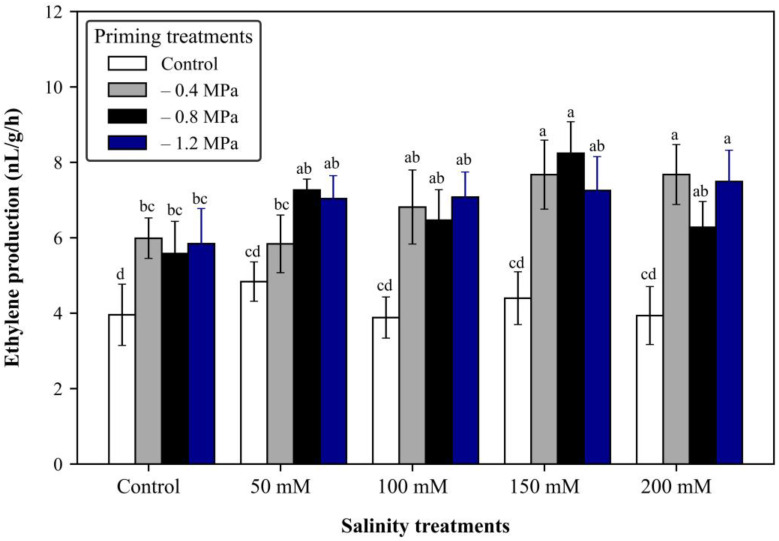
Effects of seed priming on fruit ethylene production of tomato plants under salt stress conditions. Values are expressed as mean ± SD of 10 replications (*n* = 10), and one fruit was considered as a replication. The values with the same alphabetical letters did not differ significantly, based on ANOVA followed by the Tukey test at 5%.

**Figure 3 plants-12-02187-f003:**
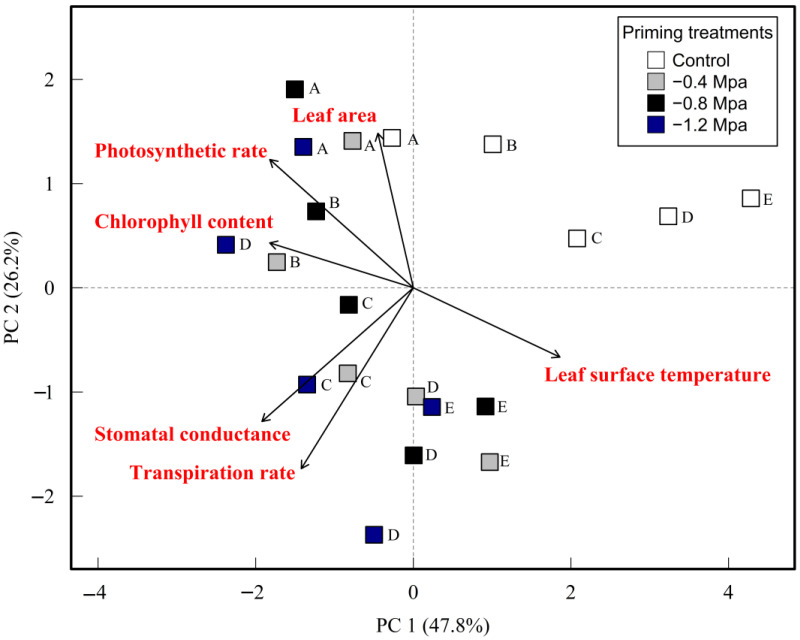
Principal component analysis of photosynthesis parameters. The letters show salinity treatments; (A) 0 mM NaCl, (B) 50 mM NaCl, (C) 100 mM NaCl, (D) 150 mM NaCl, (E) 200 mM NaCl.

**Figure 4 plants-12-02187-f004:**
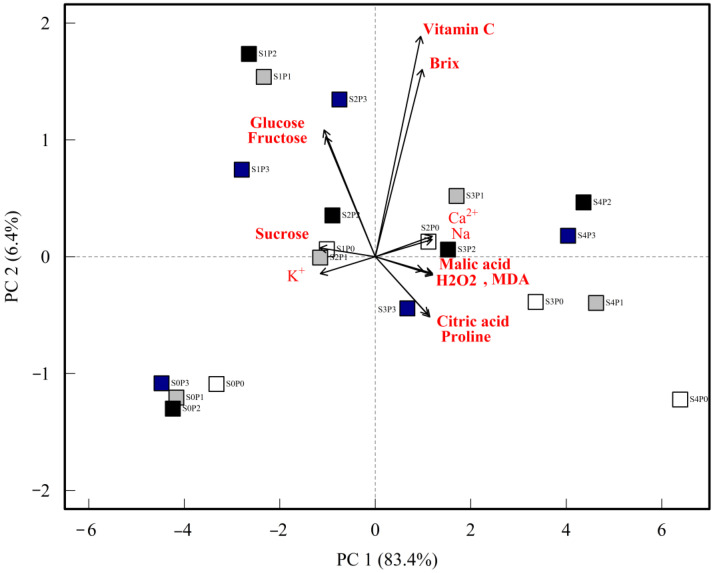
Principal component analysis of biochemical parameters.

**Figure 5 plants-12-02187-f005:**
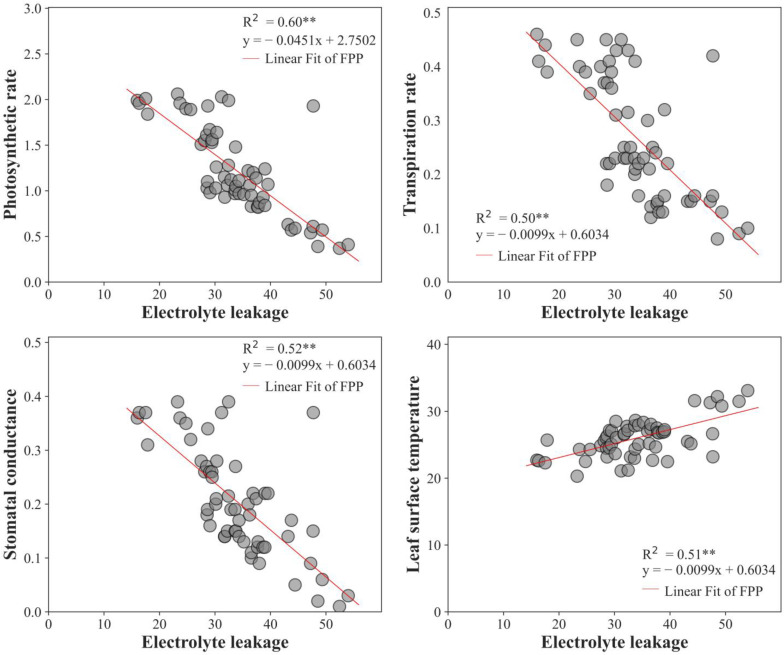
Regression analysis of photosynthetic rate, transpiration rate, and stomatal conductance with electrolyte leakage. The asterisks represent the significance. ** *p* < 0.01.

**Figure 6 plants-12-02187-f006:**
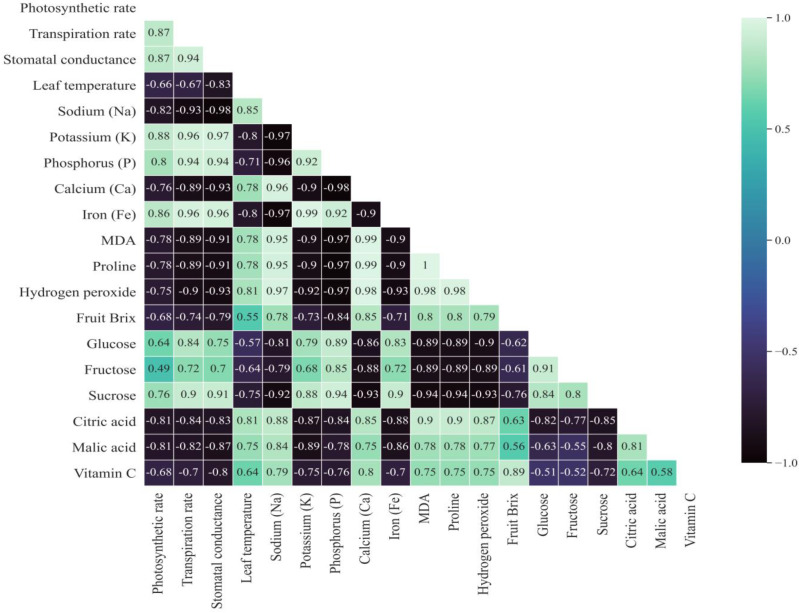
Correlation of physiological parameters with biochemical parameters.

**Table 1 plants-12-02187-t001:** Effect of priming on growth parameters of tomato plants under salt stress.

Treatments	Plant Height (cm)	Root Length (cm)	Leaves Per Plant	Leaf Length (cm)	Leaf Area (cm^2^)	Branches Per Plant
S0P0	13.85 ± 1.81 b–d	12.70 ± 0.46 b	26.67 ± 2.07 ab	6.28 ± 0.47 ab	5.66 ± 0.25 b–d	14.00 ± 1.67 b
S0P1	15.25 ± 1.80 a	14.20 ± 0.95 ab	24.00 ± 2.97 a–d	6.40 ± 0.55 ab	5.57 ± 0.25 b–e	15.50 ± 3.73 ab
S0P2	14.83 ± 1.33 ab	16.15 ± 0.15 a	28.17 ± 3.06 a	6.62 ± 0.49 a	6.36 ± 0.25 ab	15.50 ± 3.45 a
S0P3	14.27 ± 1.56 a–c	17.93 ± 0.15 a	25.83 ± 3.60 a–c	6.85 ± 0.52 a	5.17 ± 0.25 d–g	17.17 ± 4.49 a
S1P0	12.53 ± 1.17 b–e	11.60 ± 0.75 bc	21.71 ± 2.45 c–f	4.66 ± 0.24 b–d	6.00 ± 0.31 bc	8.00 ± 0.89 c–f
S1P1	11.17 ± 0.99 e–h	13.53 ± 0.45 ab	22.84 ± 1.74 b–e	5.16 ± 0.40 b	5.93 ± 0.27 b–d	9.00 ± 0.89 c–e
S1P2	12.70 ± 1.31 a–e	13.53 ± 0.55 ab	22.98 ± 2.04 b–e	4.77 ± 0.26 bc	6.95 ± 0.31 a	10.67 ± 0.82 b–d
S1P3	14.12 ± 1.52 a–d	12.30 ± 0.72 b	22.20 ± 2.40 c–f	4.55 ± 0.27 b–e	7.14 ± 0.27 a	11.17 ± 1.17 bc
S2P0	10.98 ± 0.70 e–h	9.10 ± 0.20 d	18.14 ± 3.12 f–h	3.93 ± 0.19 e–g	4.14 ± 0.25 hi	7.83 ± 0.75 c–f
S2P1	11.93 ± 1.00 c–f	10.67 ± 0.57 bc	19.89 ± 1.75 d–h	4.14 ± 0.28 c–f	3.71 ± 0.25 i	8.17 ± 0.75 c–f
S2P2	12.23 ± 1.25 c–f	11.00 ± 1.00 bc	20.60 ± 1.35 d–g	3.97 ± 0.17 d–g	4.77 ± 0.25 e–h	8.83 ± 0.75 c–e
S2P3	13.25 ± 1.14 a–d	11.80 ± 0.20 b	22.91 ± 0.97 b–e	4.13 ± 0.21 c–f	4.39 ± 0.25 g–i	9.17 ± 0.75 c–e
S3P0	9.75 ± 0.99 gh	8.73 ± 0.68 d-g	17.47 ± 1.42 gh	3.91 ± 0.15 e–g	5.51 ± 0.31 c–f	7.00 ± 0.89 ef
S3P1	10.57 ± 1.58 f–h	10.37 ± 0.65 bc	19.07 ± 0.90 e–h	4.14 ± 0.34 c–f	5.79 ± 0.27 b–d	7.67 ± 0.82 c–f
S3P2	11.58 ± 0.74 d–g	9.27 ± 0.31 cd	19.65 ± 0.94 e–h	3.38 ± 0.24 g	4.15 ± 0.31 hi	7.50 ± 0.84 def
S3P3	11.18 ± 1.28 e–h	9.93 ± 0.60 cd	18.06 ± 1.39 f–h	3.66 ± 0.31 fg	3.72 ± 0.27 i	8.00 ± 0.89 c–f
S4P0	8.85 ± 0.84 h	7.53 ± 0.68 e	16.11 ± 0.96 h	3.57 ± 0.19 gh	5.72 ± 0.25 b–d	5.17 ± 0.75 f
S4P1	11.22 ± 0.81 e–h	8.10 ± 0.62 de	18.22 ± 1.38 f–h	3.83 ± 0.18 fg	4.02 ± 0.25 hi	6.33 ± 0.52 ef
S4P2	10.58 ± 0.91 f–h	7.90 ± 0.20 de	17.25 ± 1.15 gh	3.64 ± 0.41 fg	4.73 ± 0.25 f–h	7.00 ± 0.89 ef
S4P3	10.23 ± 0.65 f–h	8.77 ± 0.64 de	17.97 ± 1.17 f–h	3.81 ± 0.35 fg	5.96 ± 0.25 b–d	8.00 ± 0.89 c–f
S	**	***	***	***	***	***
P	*	**	***	**	***	***
S×P	*	*	*	*	*	*

Data are expressed as mean ± SD of 10 replications (*n* = 10), and one plant was considered as a replication. The values with the same alphabetical letters did not differ significantly, based on ANOVA followed by the Tukey test at 5%. S stands for salinity and P for priming. *** *p* < 0.001, ** *p* < 0.01, * *p* < 0.05.

**Table 2 plants-12-02187-t002:** Effect of seed priming on the fresh and dry weight of root, shoot, and leaf of tomato plants under salt stress.

Treatments	Root		Shoot		Leaf	
	FW (g)	DW (g)	FW (g)	DW (g)	FW (g)	DW (g)
S0P0	0.53 ± 0.08 a–d	0.24 ± 0.04 b–d	7.07 ± 1.32 a	1.35 ± 0.31 a	0.48 ± 0.01 a	0.05 ± 0.01 a–d
S0P1	0.63 ± 0.07 ab	0.27 ± 0.07 ab	5.36 ± 0.99 b–d	1.07 ± 0.26 a–c	0.46 ± 0.01 ab	0.04 ± 0.01 b–e
S0P2	0.57 ± 0.03 a–c	0.21 ± 0.04 a–e	4.58 ± 0.27 b–f	0.74 ± 0.09 b–f	0.43 ± 0.01 cd	0.05 ± 0.01 a–c
S0P3	0.74 ± 0.06 a	0.30 ± 0.04 a	5.40 ± 1.31 b–d	0.91 ± 0.30 b–e	0.40 ± 0.01 ef	0.06 ± 0.02 a
S1P0	0.51 ± 0.25 b–e	0.27 ± 0.07 ab	4.04 ± 0.52 b–f	0.86 ± 0.12 b–f	0.46 ± 0.01 a–c	0.04 ± 0.01 b–d
S1P1	0.56 ± 0.05 a–c	0.25 ± 0.08 a–c	5.55 ± 1.06 a–c	1.32 ± 0.13 a	0.44 ± 0.01 bc	0.05 ± 0.01 a–d
S1P2	0.26 ± 0.18 c–f	0.12 ± 0.05 c–f	1.57 ± 0.20 e–g	0.75 ± 0.02 b–f	0.41 ± 0.01 de	0.05 ± 0.02 a–c
S1P3	0.15 ± 0.01 f	0.15 ± 0.09 b–f	1.41 ± 0.10 e–g	0.99 ± 0.09 a–d	0.38 ± 0.01 fg	0.05 ± 0.03 ab
S2P0	0.21 ± 0.09 ef	0.11 ± 0.03 c–f	3.41 ± 2.23 b–g	0.67 ± 0.06 d–g	0.37 ± 0.01 gh	0.03 ± 0.01 c–g
S2P1	0.30 ± 0.11 c–f	0.15 ± 0.05 b–f	6.26 ± 2.98 ab	1.09 ± 0.13 ab	0.36 ± 0.02 g–i	0.03 ± 0.02 c–g
S2P2	0.23 ± 0.09 d–f	0.11 ± 0.01 c–f	3.62 ± 1.08 b–f	0.71 ± 0.05 b–f	0.34 ± 0.01 i–k	0.04 ± 0.01 b–e
S2P3	0.39 ± 0.08 b–f	0.17 ± 0.04 a–f	4.96 ± 1.73 a–e	0.68 ± 0.04 c–g	0.32 ± 0.02 kl	0.04 ± 0.01 b–d
S3P0	0.24 ± 0.02 d–f	0.09 ± 0.03 ef	1.34 ± 0.16 fg	0.44 ± 0.03 g	0.35 ± 0.02 h–j	0.03 ± 0.01 d–g
S3P1	0.18 ± 0.09 ef	0.10 ± 0.04 d–f	2.28 ± 0.57 c–g	0.79 ± 0.03 b–f	0.32 ± 0.01 j–l	0.03 ± 0.02 c–g
S3P2	0.16 ± 0.05 ef	0.08 ± 0.01 ef	1.94 ± 0.62 d–g	0.59 ± 0.02 e–g	0.30 ± 0.02 lm	0.04 ± 0.01 c–g
S3P3	0.16 ± 0.04 ef	0.09 ± 0.03 ef	1.41 ± 0.59 e–g	0.77 ± 0.02 b–f	0.27 ± 0.03 n–p	0.04 ± 0.02 c–g
S4P0	0.10 ± 0.03 fg	0.06 ± 0.01 f	1.14 ± 0.43 fg	0.49 ± 0.02 fg	0.29 ± 0.01 mn	0.02 ± 0.01 g
S4P1	0.12 ± 0.07 f	0.06 ± 0.02 f	1.27 ± 1.28 fg	0.68 ± 0.13 c–g	0.27 ± 0.01 m–o	0.03 ± 0.03 fg
S4P2	0.08 ± 0.01 fg	0.05 ± 0.01 f	0.84 ± 0.47 g	0.63 ± 0.02 d–g	0.26 ± 0.01 op	0.02 ± 0.01 g
S4P3	0.13 ± 0.02 ef	0.06 ± 0.01 f	0.88 ± 0.05 g	0.53 ± 0.01 e–g	0.24 ± 0.01 p	0.03 ± 0.01 e–g
S	***	**	**	**	***	***
P	*	*	*	*	*	*
S×P	**	*	**	*	**	*

Data are expressed as mean ± SD of 10 replications (*n* = 10), and one plant was considered as a replication. The values with the same alphabetical letters did not differ significantly, based on ANOVA followed by the Tukey test at 5%. *** *p* < 0.001, ** *p* < 0.01, * *p* < 0.05. S: salinity, P: priming, PEG: Polyethylene Glycol, FW: Fresh Weight, DW: Dry Weight.

**Table 3 plants-12-02187-t003:** Effect of tomato seeds priming with PEG6000 on photosynthetic traits under salt stress.

Treatments	P_n_ (μmol CO_2_ m^−2^ s^−1^)	E (μmol CO_2_ m^−2^ s^−1^)	gs (mol CO_2_ m^−2^ s^−1^)	LT (°C)
S0P0	5.89 ± 0.05 b	0.37 ± 0.02 c	0.32 ± 0.02 b	25.44 ± 1.04 c–e
S0P1	5.93 ± 0.03 ab	0.40 ± 0.02 bc	0.36 ± 0.01 ab	23.33 ± 0.91 ef
S0P2	5.99 ± 0.03 ab	0.43 ± 0.02 ab	0.38 ± 0.01 a	22.03 ± 0.74 f
S0P3	6.03 ± 0.04 a	0.45 ± 0.01 a	0.37 ± 0.02 a	21.37 ± 1.22 f
S1P0	5.24 ± 0.02 e	0.31 ± 0.01 d	0.21 ± 0.01 d	27.59 ± 0.79 bc
S1P1	5.55 ± 0.02 cd	0.37 ± 0.02 c	0.26 ± 0.01 c	25.89 ± 1.04 b–e
S1P2	5.51 ± 0.03 d	0.40 ± 0.01 bc	0.27 ± 0.01 c	25.19 ± 1.04 c–e
S1P3	5.64 ± 0.03 c	0.43 ± 0.02 ab	0.27 ± 0.01 c	25.55 ± 0.86 b–e
S2P0	4.60 ± 0.03 l	0.15 ± 0.01 f	0.15 ± 0.02 ef	25.77 ± 0.78 b–e
S2P1	5.06 ± 0.04 f–h	0.21 ± 0.03 e	0.19 ± 0.01 de	24.01 ± 1.04 d–f
S2P2	5.02 ± 0.05 g–i	0.22 ± 0.01 e	0.20 ± 0.02 d	23.31 ± 1.04 ef
S2P3	5.15 ± 0.04 ef	0.25 ± 0.01 e	0.21 ± 0.02 d	23.51 ± 1.04 ef
S3P0	4.57 ± 0.03 l	0.15 ± 0.02 f	0.07 ± 0.02 h	31.23 ± 0.40 a
S3P1	5.01 ± 0.05 g–i	0.21 ± 0.02 e	0.14 ± 0.01 fg	28.28 ± 0.40 b
S3P2	5.98 ± 0.05 hi	0.23 ± 0.01 e	0.15 ± 0.01 ef	27.25 ± 0.66 bc
S3P3	5.11 ± 0.05 fg	0.23 ± 0.02 e	0.15 ± 0.02 ef	26.45 ± 1.35 b–d
S4P0	4.39 ± 0.02 m	0.09 ± 0.01 g	0.02 ± 0.01 i	32.27 ± 0.80 a
S4P1	4.85 ± 0.02 jk	0.13 ± 0.02 fg	0.11 ± 0.02 gh	27.49 ± 0.81 bc
S4P2	4.82 ± 0.02 k	0.15 ± 0.01 f	0.12 ± 0.01 fg	26.83 ± 0.15 bc
S4P3	4.95 ± 0.02 ij	0.14 ± 0.02 f	0.12 ± 0.02 fg	27.63 ± 0.67 bc
S	***	***	***	***
P	**	*	**	**
S×P	***	**	**	***

Data are expressed as mean ± SD of 10 replications (*n* = 10), and one plant was considered as a replication. The values with the same alphabetical letters did not differ significantly, based on ANOVA followed by the Tukey test at 5%. *** *p* < 0.001, ** *p* < 0.01, * *p* < 0.05. S: salinity, P: priming, P_n_: Photosynthetic rate, E: Transpiration rate, gs: Stomatal conductance, LT: leaf surface temperature.

**Table 4 plants-12-02187-t004:** Effects of seed priming on fruit yield attributes of tomato plants under salt stress conditions.

Treatments	Fruit Size (mm)	Fruit Weight (g)	Fruits Per Plant	Yield Per Plant (g)
S0P0	19.6 ± 1.1 b–d	9.04 ± 2.00 a	18.17 ± 4.17 a–c	63.45 ± 13.68 a–c
S0P1	19.4 ± 0.9 cd	8.84 ± 2.05 a	21.00 ± 3.58 ab	67.01 ± 11.90 ab
S0P2	19.0 ± 1.8 cd	8.44 ± 2.33 ab	21.00 ± 3.63 ab	73.87 ± 13.93 a
S0P3	20.7 ± 0.9 a	8.07 ± 1.74 a–c	22.33 ± 5.96 a	74.61 ± 20.58 a
S1P0	17.5 ± 0.8 d–h	6.12 ± 1.82 a–d	18.67 ± 1.21 a–c	41.96 ± 13.71 c–f
S1P1	18.8 ± 0.8 b–e	6.01 ± 1.88 a–d	20.17 ± 1.83 ab	42.78 ± 25.72 c–f
S1P2	18.3 ± 0.8 c–f	5.75 ± 1.83 b–e	21.33 ± 1.63 a	44.50 ± 14.24 b–e
S1P3	22.2 ± 1.0 a	5.05 ± 1.41 c–f	22.00 ± 1.41 a	47.34 ± 24.07 b–d
S2P0	15.3 ± 0.8 h–j	4.91 ± 2.33 d–f	16.50 ± 1.05 b–e	33.74 ± 0.84 d–g
S2P1	16.5 ± 0.8 e–h	4.87 ± 2.42 d–f	17.67 ± 1.51 a–d	28.65 ± 2.67 d–g
S2P2	16.0 ± 0.6 f–i	4.47 ± 0.84 d–g	18.17 ± 1.60 a–c	35.00 ± 1.93 d-e
S2P3	17.7 ± 0.5 d–f	3.15 ± 0.59 d–h	18.17 ± 0.75 a–c	34.95 ± 2.42 d–g
S3P0	12.2 ± 0.5 kl	2.82 ± 0.43 e–h	11.17 ± 0.75 fg	18.73 ± 2.25 fg
S3P1	14.0 ± 0.9 i–k	2.70 ± 0.51 f–h	13.00 ± 0.63 d–g	21.95 ± 3.64 e–g
S3P2	12.8 ± 0.8 kl	2.45 ± 0.55 f–h	12.67 ± 0.82 e–g	20.09 ± 1.21 fg
S3P3	15.7 ± 0.8 gh	2.15 ± 0.05 f–h	14.17 ± 1.47 c–g	24.29 ± 1.91 d–g
S4P0	11.2 ± 1.0 l	2.03 ± 0.23 f–h	9.83 ± 1.17 g	16.78 ± 1.03 g
S4P1	12.2 ± 0.8 kl	1.62 ± 0.07 gh	12.50 ± 0.55 e–g	17.00 ± 1.79 g
S4P2	13.3 ± 0.8 jk	1.20 ± 0.13 h	14.00 ± 0.89 c–g	17.72 ± 4.14 g
S4P3	11.7 ± 0.8 kl	1.18 ± 0.10 h	15.33 ± 1.03 c–f	19.77 ± 1.53 fg
S	**	**	***	***
P	*	Ns	***	**
S×P	*	*	*	*

Data are expressed as mean ± SD of 10 replications (*n* = 10), and one plant was considered as a replication. The values with the same alphabetical letters did not differ significantly, based on ANOVA followed by the Tukey test at 5%. S: salinity, P: priming, Ns: not significant, *** *p* < 0.001, ** *p* < 0.01, * *p* < 0.05.

**Table 5 plants-12-02187-t005:** Effects of seed priming on fruit quality of tomato plants under salt stress conditions.

Treatments	Fruit Color (* a Value)	Fruit Brix (%)	Glucose $\left(mg\times g^{-1}\;FW\right)$	Fructose $\left(mg\times g^{-1}\;FW\right)$	Sucrose $\left(mg\times g^{-1}\;FW\right)$	Citric Acid $\left(mg\times g^{-1}\;FW\right)$	Malic Acid $\left(mg\times g^{-1}\;FW\right)$	Vitamin C $\left(mg100g^{-1}\;FW\right)$
S0P0	46.54 ± 2.01 bc	4.20 ± 0.46 i	51.19 ± 3.56 ab	72.67 ± 2.50 de	16.05 ± 1.59 ab	5.47 ± 0.14 d–f	3.03 ± 0.31 b–d	23.83 ± 1.89 f–h
S0P1	55.54 ± 2.01 b	4.65 ± 0.25 hi	46.41 ± 2.72 b–d	76.57 ± 3.48 cd	16.61 ± 1.52 a	5.04 ± 0.13 ef	2.48 ± 0.42 d	22.27 ± 1.53 gh
S0P2	46.75 ± 1.73 bc	4.83 ± 0.68 g–i	46.38 ± 2.77 b–d	77.82 ± 2.80 b–d	14.26 ± 0.72 b–d	4.75 ± 0.24 f	2.37 ± 0.48 d	21.27 ± 1.42 h
S0P3	41.55 ± 1.57 c	4.73 ± 0.21 g–i	47.34 ± 3.22 bc	79.41 ± 1.88 a–d	16.87 ± 1.56 a	4.71 ± 0.17 f	2.65 ± 0.21 cd	21.93 ± 0.65 h
S1P0	36.52 ± 1.86 d	6.93 ± 1.00 f–i	51.74 ± 3.39 ab	82.08 ± 3.07 a–c	13.64 ± 1.12 b–e	6.37 ± 0.44 c–e	3.88 ± 0.27 a–c	26.80 ± 2.16 d–g
S1P1	68.53 ± 1.15 ab	8.10 ± 1.00 d–f	57.45 ± 2.14 a	84.80 ± 2.27 ab	14.32 ± 1.05 a–d	5.36 ± 0.35 ef	2.70 ± 0.41 cd	30.80 ± 0.20 a–d
S1P2	58.53 ± 1.15 b	8.23 ± 0.42 c–f	56.45 ± 1.84 a	85.70 ± 3.42 a	15.75 ± 1.31 ab	5.05 ± 0.35 ef	2.65 ± 0.29 cd	31.40 ± 2.80 a–d
S1P3	51.30 ± 1.76 bc	7.40 ± 0.44 d–g	57.48 ± 2.54 a	79.66 ± 3.35 a–d	15.20 ± 2.11 a–c	4.69 ± 0.49 f	3.41 ± 0.46 b–d	27.33 ± 1.85 c–f
S2P0	42.17 ± 1.01 c	7.43 ± 0.71 d–g	41.31 ± 2.62 c–f	77.64 ± 1.91 b–d	12.62 ± 0.67 b–e	6.72 ± 0.54 b–d	4.06 ± 0.82 ab	31.00 ± 1.40 a–d
S2P1	40.32 ± 1.10 c	8.00 ± 0.95 d–f	46.99 ± 1.09 bc	79.95 ± 2.02 a–d	13.86 ± 0.94 b–e	5.26 ± 0.33 ef	3.31 ± 0.35 b–d	25.00 ± 0.20 e–h
S2P2	71.26 ± 1.06 a	7.23 ± 0.50 e–h	44.67 ± 0.96 b–e	79.69 ± 1.85 a–d	13.69 ± 1.06 a–d	5.64 ± 0.37 d–f	3.43 ± 0.5 b–d	28.93 ± 1.47 b–e
S2P3	61.26 ± 1.06 b	10.13 ± 0.42 a–c	44.88 ± 2.38 b–e	81.64 ± 1.92 a–c	13.39 ± 1.16 a–d	5.34 ± 0.34 ef	3.60 ± 0.52 a–d	30.33 ± 1.03 b–d
S3P0	38.67 ± 3.59 d	9.17 ± 1.45 c–f	34.16 ± 2.05 fg	63.52 ± 2.08 f	11.97 ± 0.23 c–f	7.89 ± 0.22 ab	4.07 ± 0.52 ab	30.53 ± 0.83 b–d
S3P1	51.16 ± 0.87 bc	9.70 ± 2.21 a–d	38.53 ± 1.11 ef	67.71 ± 2.40 ef	12.80 ± 1.58 b–e	5.82 ± 0.37 d–f	3.30 ± 0.42 b–d	31.47 ± 0.64 a–c
S3P2	45.32 ± 1.69 bc	8.47 ± 0.99 c–f	39.62 ± 1.54 d–f	62.32 ± 1.65 f	11.87 ± 1.31 c–f	5.52 ± 0.46 d–f	3.52 ± 0.34 a–d	30.87 ± 1.40 a–d
S3P3	74.57 ± 1.14 a	7.70 ± 0.72 d–f	40.64 ± 3.56 c–f	62.16 ± 2.09 f	12.65 ± 0.61 b–e	5.44 ± 0.21 d–f	3.27 ± 0.20 b–d	28.20 ± 1.59 c–f
S4P0	44.57 ± 1.14 c	8.97 ± 0.57 b–f	26.76 ± 2.09 h	49.57 ± 2.05 g	8.59 ± 0.46 f	8.62 ± 0.39 a	4.73 ± 0.26 a	31.93 ± 1.47 a–c
S4P1	58.22 ± 2.15 b	10.97 ± 0.38 a–c	28.64 ± 1.52 gh	52.16 ± 1.86 g	11.06 ± 0.91 d–f	7.36 ± 0.69 bc	4.11 ± 0.21 ab	30.73 ± 1.50 a–d
S4P2	55.27 ± 0.60 b	11.53 ± 0.57 ab	28.39 ± 0.71 gh	49.24 ± 3.75 g	11.02 ± 1.01 d–f	7.55 ± 0.62 bc	3.49 ± 0.33 a–d	35.33 ± 0.42 a
S4P3	50.28 ± 1.86 bc	11.77 ± 1.19 a	29.37 ± 1.98 gh	47.97 ± 2.05 g	10.53 ± 0.64 ef	7.46 ± 0.94 bc	3.59 ± 0.35 a–d	33.00 ± 2.23 ab
S	**	***	***	***	***	***	***	***
P	*	*	**	*	*	***	***	*
S×P	*	**	*	*	*	*	*	***

Data are expressed as mean ± SD of 10 replications (*n* = 10), and one fruit was considered as a replication. The values with the same alphabetical letters did not differ significantly, based on ANOVA followed by the Tukey test at 5%. S: salinity, P: priming. *** *p* < 0.001, ** *p* < 0.01, * *p* < 0.05.

**Table 6 plants-12-02187-t006:** Leaf MDA, proline, and hydrogen peroxide content in response to salt stress.

Treatments	MDA $\left(nmol\;g^{-1}\;FW\right)$	Proline $\left(\mu mol\;g^{-1}\;FW\right)$	H_2_O_2_ $\left(\mu mol\;g^{-1}\;FW\right)$	Electrolyte Leakage (%)
S0P0	4.45 ± 0.47 kl	3.64 ± 0.16 f–h	6.21 ± 0.96 k–m	26.08 ± 2.31 f–h
S0P1	3.93 ± 0.17 l	3.37 ± 0.18 h	5.26 ± 0.24 lm	24.15 ± 1.93 h
S0P2	3.72 ± 0.36 l	3.39 ± 0.17 h	4.74 ± 0.37 m	28.18 ± 2.65 d–f
S0P3	3.54 ± 0.44 l	3.23 ± 0.12 h	4.74 ± 0.53 m	24.82 ± 2.39 gh
S1P0	7.74 ± 0.64 g–i	4.23 ± 0.29 fg	12.23 ± 2.49 hi	36.06 ± 2.48 c–e
S1P1	6.76 ± 0.19 h–j	3.65 ± 0.07 f–h	9.14 ± 0.82 i–k	27.27 ± 1.79 e–g
S1P2	6.03 ± 0.06 i–k	3.56 ± 0.06 gh	8.69 ± 0.54 jk	30.27 ± 2.10 d–f
S1P3	5.76 ± 0.40 jk	3.51 ± 0.08 gh	8.31 ± 0.43 j–l	27.71 ± 1.09 e–g
S2P0	9.84 ± 0.83 ef	5.46 ± 0.41 b–d	18.31 ± 1.20 g	44.31 ± 1.35 a–c
S2P1	8.04 ± 0.75 gh	4.34 ± 0.29 ef	13.30 ± 0.84 h	32.40 ± 1.73 d–f
S2P2	8.47 ± 0.48 f–h	3.89 ± 0.17 f–h	11.67 ± 0.57 h–j	34.83 ± 1.99 c–e
S2P3	8.60 ± 0.45 fg	3.69 ± 0.24 f–h	12.35 ± 0.57 hi	35.81 ± 1.24 c–e
S3P0	15.09 ± 0.22 c	5.85 ± 0.25 bc	27.10 ± 1.04 cd	48.91 ± 3.77 ab
S3P1	11.83 ± 0.64 d	5.66 ± 0.15 b–d	24.51 ± 1.17 de	33.97 ± 0.79 c–e
S3P2	11.52 ± 0.53 de	5.45 ± 0.09 b–d	23.60 ± 0.59 ef	32.22 ± 0.94 d–f
S3P3	11.82 ± 0.64 d	5.08 ± 0.06 de	20.77 ± 1.16 fg	33.61 ± 1.06 d–f
S4P0	22.29 ± 0.82 a	7.07 ± 0.72 a	35.93 ± 1.59 a	50.36 ± 1.38 a
S4P1	19.40 ± 0.99 b	5.90 ± 0.12 b	32.90 ± 1.51 ab	36.23 ± 1.12 c–e
S4P2	18.33 ± 0.71 b	5.36 ± 0.21 b–d	30.36 ± 1.40 bc	38.53 ± 0.94 b–d
S4P3	17.75 ± 0.46 b	5.12 ± 0.13 cd	28.80 ± 1.13 c	37.04 ± 0.86 c–e
S	***	***	***	***
P	***	***	***	**
S×P	**	***	**	***

Data are expressed as mean ± SD of 10 replications (*n* = 10). The values with the same alphabetical letters did not differ significantly, based on ANOVA followed by the Tukey test at 5%. S: salinity, P: priming. *** *p* < 0.001, ** *p* < 0.01.

**Table 7 plants-12-02187-t007:** Effects of seed priming on nutrient accumulation in tomato leaves under salt stress conditions.

Treatments	$Na^{+}$ $\left(mg\times g^{-1}\;DW\right)$	$K^{+}$ $\left(mg\times g^{-1}\;DW\right)$	P $\left(mg\times g^{-1}\;DW\right)$	$Ca^{2+}$ $\left(mg\times g^{-1}\;DW\right)$	Mg $\left(mg\times g^{-1}\;DW\right)$	Fe $\left(mg\times g^{-1}\;DW\right)$	Zn $\left(mg\times g^{-1}\;DW\right)$
S0P0	0.23 ± 0.006 i	31.52 ± 1.01 bc	8.46 ± 0.03 c	1.55 ± 0.015 m	5.73 ± 0.012 c	0.25 ± 0.011 ab	0.019 ± 0.002 h
S0P1	0.19 ± 0.01 j	34.56 ± 0.77 ab	8.63 ± 0.04 a	1.44 ± 0.010 n	5.83 ± 0.014 b	0.26 ± 0.015 ab	0.008 ± 0.002 ij
S0P2	0.16 ± 0.005 j	36.26 ± 1.28 a	8.56 ± 0.06 ab	1.38 ± 0.016 o	5.88 ± 0.012 a	0.28 ± 0.013 ab	0.006 ± 0.001 jk
S0P3	0.17 ± 0.008 j	34.51 ± 1.30 ab	8.52 ± 0.03 bc	1.34 ± 0.018 o	5.88 ± 0.009 a	0.28 ± 0.011 a	0.004 ± 0.001 k
S1P0	0.32 ± 0.008 f	27.55 ± 0.78 de	7.26 ± 0.03 e	1.94 ± 0.019 j	5.01 ± 0.012 e	0.20 ± 0.009 cd	0.039 ± 0.003 d
S1P1	0.27 ± 0.002 gh	31.22 ± 1.15 bc	7.99 ± 0.03 d	1.84 ± 0.013 k	5.09 ± 0.015 d	0.24 ± 0.010 bc	0.025 ± 0.002 g
S1P2	0.26 ± 0.004 g–i	30.61 ± 0.85 cd	7.97 ± 0.03 d	1.81 ± 0.021 kl	5.12 ± 0.013 d	0.26 ± 0.011 ab	0.020 ± 0.001 h
S1P3	0.24 ± 0.003 hi	32.54 ± 1.08 bc	7.99 ± 0.03 d	1.79 ± 0.014 l	5.13 ± 0.015 d	0.27 ± 0.013 ab	0.010 ± 0.001 i
S2P0	0.37 ± 0.075 e	21.53 ± 0.92 hi	6.38 ± 0.03 h	2.13 ± 0.017 g	4.71 ± 0.013 g	0.12 ± 0.012 gh	0.048 ± 0.003 c
S2P1	0.32 ± 0.064 f	26.34 ± 0.96 ef	6.53 ± 0.02 g	2.05 ± 0.015 h	4.80 ± 0.014 f	0.18 ± 0.011 d–f	0.031 ± 0.003 f
S2P2	0.32 ± 0.008 f	25.61 ± 1.14 e-g	6.61 ± 0.02 fg	2.01 ± 0.015 hi	4.81 ± 0.015 f	0.18 ± 0.011 de	0.025 ± 0.022 g
S2P3	0.28 ± 0.010 g	27.55 ± 1.22 de	6.62 ± 0.03 f	1.99 ± 0.017 ij	4.84 ± 0.008 f	0.20 ± 0.009 cd	0.018 ± 0.001 h
S3P0	0.46 ± 0.006 b	19.57 ± 1.13 ij	5.35 ± 0.02 j	2.61 ± 0.016 d	4.15 ± 0.009 i	0.08 ± 0.008 i	0.052 ± 0.004 b
S3P1	0.40 ± 0.005 de	22.57 ± 1.34 g–i	5.85 ± 0.02 i	2.53 ± 0.013 e	4.24 ± 0.009 h	0.13 ± 0.012 gh	0.035 ± 0.004 e
S3P2	0.40 ± 0.008 de	24.65 ± 0.88 e–h	5.82 ± 0.03 i	2.48 ± 0.012 ef	4.26 ± 0.011 h	0.15 ± 0.014 e–g	0.031 ± 0.003 f
S3P3	0.37 ± 0.006 e	26.69 ± 1.12 ef	5.80 ± 0.02 i	2.47 ± 0.011 f	4.27 ± 0.010 h	0.19 ± 0.013 de	0.025 ± 0.003 g
S4P0	0.52 ± 0.004 a	16.62 ± 0.85 j	4.18 ± 0.03 l	3.23 ± 0.014 a	3.72 ± 0.015 l	0.06 ± 0.011 i	0.062 ± 0.005 a
S4P1	0.46 ± 0.007 b	19.58 ± 1.08 ij	4.36 ± 0.02 k	3.11 ± 0.018 b	3.83 ± 0.012 k	0.10 ± 0.015 hi	0.041 ± 0.004 d
S4P2	0.45 ± 0.008 bc	22.43 ± 1.28 g–i	4.40 ± 0.03 k	3.06 ± 0.017 bc	3.85 ± 0.013 jk	0.13 ± 0.011 gh	0.031 ± 0.004 f
S4P3	0.42 ± 0.009 cd	23.62 ± 1.20 f–h	4.39 ± 0.02 k	3.05 ± 0.021 c	3.88 ± 0.013 j	0.14 ± 0.013 fg	0.027 ± 0.003 g
S	***	***	***	***	***	***	***
P	***	***	***	***	***	***	***
S×P	*	**	***	*	*	*	***

Data are expressed as mean ± SD of 10 replications (*n* = 10), and one plant was considered as a replication. The values with the same alphabetical letters did not differ significantly, based on ANOVA followed by the Tukey test at 5%. S: salinity, P: priming. *** *p* < 0.001, ** *p* < 0.01, * *p* < 0.05.

**Table 8 plants-12-02187-t008:** Combination of salinity and PEG seed pre-sowing priming treatments.

Abbreviations	Salinity	Priming
S0P0	0 mM	0 MPa
S0P1	0 mM	−0.4 MPa
S0P2	0 mM	−0.8 MPa
S0P3	0 mM	−1.2 MPa
S1P0	50 mM	0 MPa
S1P1	50 mM	−0.4 MPa
S1P2	50 mM	−0.8 MPa
S1P3	50 mM	−1.2 MPa
S2P0	100 mM	0 MPa
S2P1	100 mM	−0.4 MPa
S2P2	100 mM	−0.8 MPa
S2P3	100 mM	−1.2 MPa
S3P0	150 mM	0 MPa
S3P1	150 mM	−0.4 MPa
S3P2	150 mM	−0.8 MPa
S3P3	150 mM	−1.2 MPa
S4P0	200 mM	0 MPa
S4P1	200 mM	−0.4 MPa
S4P2	200 mM	−0.8 MPa
S4P3	200 mM	−1.2 MPa

## Data Availability

Not applicable.

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
