# Peer review of "Potential Benefits of Seed Priming under Salt Stress Conditions on Physiological, and Biochemical Attributes of Micro-Tom Tomato Plants"

_plants, 2023, doi:10.3390/plants12112187_

Round 1
Reviewer 1 Report
The MS entitle; Investigating the potential benefits of seed priming under salt stress conditions on physiology, and biochemical attributes of Micro-Tom tomato plants.
I recommend minor revision and suggest the following point need to improve before accepting.
Due to a lot of processing in this experiment design, the workload of this experiment is relatively large, and a lot of growth and physiological indexes are also measured. However, the following problems need to be modified in this paper:
1. It is suggested that the secondary heading of the results section be more specific, so that people can know more quickly what kind of treatment has what kind of impact.
2. What does the "-" between the significances in the table mean? For example, what does the "-" in "b-d" in Table 1 mean? I think it is necessary to explain. The last three lines in the table do not quite understand what you want to express.
3. Is "Fruit size" in Table
4 the transverse or longitudinal diameter of the fruit? 5. What is the meaning of "Fruit color a" in Table 5?
6. Is “-”omitted in the legend of Figure 2, 3 and 4? Because it is different from the -0.4Mpa, -0.8Mpa and -1.2Mpa mentioned in the article. Finally, I would like to ask a question: Why is Mrico-Tom used as research material? It is mentioned in the paper that the ultimate goal is for commercial application. I think it is more valuable and in line with the purpose of research to use the main popular local cultivar as the research material.
This MS needed to improve English language.
Needs to improve English.
Author Response
Thank you for your precious time and valuable comments. Please kindly refer to the attached file.

Reviewer 2 Report
1. The "Abstract" should be concise.
2. The experiment methods should be provided in detial. For instance, were the treatments started on which stage, how often and how long time did the treatments were carried out?
3. Comparing figures, such as displaying plants' height of different treatments, should be provided (if possible).
Minor editing of English language required
Author Response
We appreciate your valuable comments and suggestion. Please refer to the attached file.

Reviewer 3 Report
According to the study of the manuscript with the number “plants-2426446” and tittle “Investigating the potential benefits of seed priming under salt stress conditions on physiology, and biochemical attributes of Micro-Tom tomato plants” I did not find any major loopholes so the manuscript can be accepted in its present form.
Author Response
Thank you very much for your comment and great feedback about our article.

Reviewer 4 Report
In my opinion, the research carried out is interesting, as it combines the pre-priming of the seeds with different levels of salt stress; being quite complete at the level of biochemical and physiological parameters studied. The results validate the conclusions obtained. Although at the methodological level it seems to be well executed, when translating it into the manuscript there are many details that must be clarified, and it should be included details of the agronomic conditions in which the study was carried out. Although it is well written, most of the time comparisons are indicated at general level and not quantitatively. Statistically should be better described, and if possible indicating details of the ANOVA's (F, df, p). The discussion has to be better referenced with bibliography in some parts of it. The format of the manuscript should be reviewed. In my opinion, the investigation should be published once the issues indicated in the attached file have been clarified or corrected. English is quite good, but if possible, a native English speaker should review it to get a better fluent writing.

English is quite good, but if possible, a native English speaker should review it to get a better fluent writing.
Author Response
We the authors really appreciate your comments and suggestions for improving the quality of our manuscript.
Please kindly refer to the attached file.
